

# Iron overload induces islet β cell ferroptosis by activating ASK1/P-P38/CHOP signaling pathway

Ling Deng[1], Man-Qiu Mo[2], Jinling Zhong[1], Zhengming Li[3], Guoqiao Li[1] and Yuzhen Liang[1]

[1] Department of Endocrinology, The Second Affiliated Hospital of Guangxi Medical University, Nanning, China
[2] Department of Geriatric Endocrinology and Metabolism, The First Affiliated Hospital of Guangxi Medical University, Nanning, China
[3] Department of Endocrinology, People's Hospital of Guangxi Zhuang Autonomous Region, Nanning, China

## ABSTRACT

**Background:** Recent studies have shown that the accumulation of free iron and lipid peroxides will trigger a new form of cell death—ferroptosis. This form of cell death is associated with a variety of diseases, including type 2 diabetes. We hypothesize that iron overload may play a role in driving glucose metabolism abnormalities by inducing endoplasmic reticulum stress that mediates ferroptosis in islet β cells. In this study, we tested this conjecture from *in vivo* and *in vitro* experiments.

**Methods:** We established a mouse iron overload model by intraperitoneal injection of iron dextrose (50 mg/kg) and an iron overload cell model by treating MIN6 cells with ferric ammonium citrate (640 μmol/L, 48 h) *in vitro*. The iron deposition in pancreatic tissue was observed by Prussian blue staining, and the pathological changes in pancreatic tissues by HE staining and the protein expression level by pancreatic immunohistochemistry. In the cellular experiments, we detected the cell viability by CCK8 and observed the cellular ultrastructure by transmission electron microscopy. We also used MDA and ROS kits to detect the level of oxidative stress and lipid peroxidation in cells. Western blotting was performed to detect the expression levels of target proteins.

**Results:** Iron overload induces MIN6 cell dysfunction, leading to increased fasting blood glucose, impaired glucose tolerance, and significantly decreased insulin sensitivity in mice. This process may be related to the ferroptosis of islet β cells and the activation of ASK1/P-P38/CHOP signaling pathway.

## INTRODUCTION

Type 2 diabetes is a growing health problem, with global epidemiological findings indicating that the number of people aged 20-79 with diabetes is expected to increase to 642 million by 2040, which will carry serious implications for financial and health systems (*Ogurtsova et al., 2017*). The latest epidemiological survey of diabetes in China shows that

Corresponding author
Yuzhen Liang,
liangyuzhen26@163.com

the prevalence of diabetes is 11.2%, with type 2 diabetes accounting for more than 90% of diabetic patients, but the control rate is only 49%. The significant pathophysiological features of type 2 diabetes are known to be a decrease in the ability of insulin to regulate glucose metabolism (insulin resistance) accompanied by a decrease in insulin secretion (relative decrease) due to defective islet β cell function,however, its etiology and pathogenesis remain unclear.

As micronutrients come into the focus of research on metabolic syndrome, the link between abnormal iron metabolism and disorders of glucose metabolism has become a major research topic. Iron is the most abundant trace metal element and plays an important physiological role in the human body (*Bogdan et al., 2016*). Dietary iron ($Fe^{3+}$) is absorbed in the epithelial cells of the duodenum and transported out of the intestinal epithelium by ferroportin (FPN), which then binds to transferrin to form TF. $Fe^{3+}$ complex is transported *via* blood to the site of action (*Torti & Torti, 2013*). The regulation of iron homeostasis in the body is carried out mainly in the iron absorption pathway, which is regulated by iron-regulating factors. Once the body inputs too much iron or has abnormal iron transporter protein activity, it often leads to iron overload (*Billesbølle et al., 2020*), which further causes the development of diseases such as inflammation, neurodegeneration, tumours and endocrine metabolic diseases.

A cohort study suggests that iron overload increases the risk of type 2 diabetes, possibly due to abnormal lipid metabolism and damage to hepatocytes and duodenal epithelial cells. However, whether iron overload directly affects pancreatic islet β-cell function is still being explored (*Gao et al., 2022*). Previous studies have shown that iron overload is an important risk factor for type 2 diabetes and that excess iron has a serious negative impact on insulin secretion (*Simcox & McClain, 2013*). Iron overload causes islet dysfunction through the induction of severe oxidative stress and the production of large amounts of reactive oxygen species. However, the exact molecular mechanism remains unclear (*Marku et al., 2021*; *Hansen et al., 2012*; *Li et al., 2020*).

Ferroptosis is a novel form of cell death induced by lipid peroxide accumulation, and its occurrence is dependent on ferric ions (*Dixon et al., 2012*). On the one hand, free iron promotes the Fenton reaction and induces oxidative stress in cells, generating large amounts of oxygen free radicals and triggering a series of cell damage reactions (*Yin, Xu & Porter, 2011*). On the other hand, iron ions are involved in the synthesis of ferroptosis-related enzymes; for example, $Fe^{2+}$ acts as a cofactor for lipoxygenase to promote the peroxidation of polyunsaturated fatty acids (PUFAs), which are derived mainly from the labile iron pool (LIP) (*Philpott, Patel & Protchenko, 2020*).

Deferasirox (DFX) is a novel oral iron chelator that is commonly used clinically to treat iron overload resulting from long-term blood transfusions. Its molecular mechanism may be through inhibition of the mitogen activated protein kinase (MAPK) signalling pathway that is activated by iron overload (*Al-Rousan et al., 2009*). In contrast, GADD153, a downstream molecule of MAPK/P38, also known as DDIT3 and CHOP, is an endoplasmic reticulum stress-related factor associated with autophagy and apoptosis (*Yang et al., 2021*; *Hu et al., 2018*). Recent studies have shown that CHOP is one of the ferroptosis-inducing factors associated with PERK-eIF2α-ATF4 pathway activation mediating endoplasmic

reticulum stress (*Lee et al., 2018*). We therefore speculate that MAPK/P38- CHOP may be associated with ferroptosis induced by iron overload, while this hypothesis will be tested by *in vitro* and *in vivo* experiments.

## MATERIALS AND METHODS

### Animal experiments

All animal procedures were approved by the People's Republic of China 2021 Guide for Euthanasia of Laboratory Animals (GB/T39760-2021), the 2018 Guide for Ethical Review of Laboratory Animal Welfare (GB/T35892-2018), and the Guangxi Medical University Laboratory Animal Ethics Review Body (Project Licences: 20211007). Male C57BL/6J mice (6 weeks old) for the experiments were obtained from the Animal Experimentation Centre of Guangxi Medical University (Guangxi, China) and housed in the specific pathogen free (SPF) class animal house of the Animal Experimentation Centre of Guangxi Medical University (Guangxi, China) in accordance with standard husbandry conditions that with regular light/dark cycles and free access to water and food diet. Based on the experimental request, we randomly divided 40 mice into two groups: the control group ($n$ = 20) and the iron overload model group ($n$ = 20). The mice in the model group were given intraperitoneal injections of iron dextrose (Macklin; Shanghai, China) at a dose of 50 mg/kg twice a week for 4 weeks (*Okabe et al., 2014*). The control group was given an intraperitoneal injection of an equal volume of saline. After 4 weeks, 20 mice were divided into two subgroups. The final groups were: normal control group (control), DFX control group (DFX), iron overload group (IO) and DFX treatment group (IO+DFX).

The treatment dose of DFX was: 125 mg/kg, five times a week for 5 weeks (*Wu et al., 2018*). During this period, the body weight of the mice was monitored weekly and fasting blood glucose was monitored every fortnight. At the end of the intervention, a 2 h OGTT test was performed, and finally, serum and pancreatic tissues were collected from the mice for subsequent experiments. To avoid the effects of anesthetics on blood glucose in mice, we collected blood as quickly as possible and comforted the mice.

We consider early termination when significant weight loss (rapid onset of 15–20% of body weight), weakness, dying, or organ failure are observed. Necessary measures were used to minimize the animals suffering during the experiment. At the end of the experiment, mice were euthanized by intraperitoneal injection of chloral hydrate (10%) followed by cervical dislocation.

### Cell culture

We used the mouse islet β cell line MIN6 (purchased from Wuhan Pronosai Life Sciences Co. PNS-MC-75) as the *in vitro* study subject. All experimental cell passages were less than 30. The medium was prepared according to the following composition: 89% Dulbecco's modified Eagle's medium (DMEM) (Shanghai XP Biomed Ltd; Shanghai, China) + 10% foetal bovine serum (FBS) (Shanghai XP Biomed Ltd; Shanghai, China) + 1% cyan chain double antibodies (Shanghai XP Biomed Ltd; Shanghai, China), and the cell culture chamber (Thermo Fisher; Shanghai, China) conditions were set at 5% $CO_2$ and 37 °C. The cells were cultured using a 25 $cm^2$ permeable culture flask (Corning; New York NY,

USA). Generally, when the cell density reaches 80%, the cells can be passaged at a ratio of 1:4.

## Modelling iron overload in MIN6 cells

The intervention was performed on MIN6 cells with media containing different concentrations of ferric ammonium citrate (Yuanye Bio-Technology; Shanghai, China). Based on the results of the LIP, MIN6 cells had iron overload when there was a significant difference in LIP levels in the intervention group compared to the control group; combined with the results of cell viability assay, the intervention concentration and time with higher LIP levels and at least 50% cell viability were selected as the intervention conditions for the subsequent establishment of the iron overload model.

## Detection of cell viability

Cell viability was assayed by the Cell Counting Kit 8 (CCK8) method. Cell suspensions of 200 μl at a density of $10 * 104$/ml were inoculated in 96-well plates and incubated for 24 h. The plates were then incubated for 24, 48 and 72 h, respectively, protected from light using 20/40/80/160/320/640/1,280/2,560 μM concentrations of ferric ammonium citrate (FAC) medium. At the end of the intervention time the old medium was removed and washed with phosphate-buffered saline (PBS) once. Medium containing 10% CCK8 (Meilunbio; Dalian, China) solution was added and incubated for 30 min away from light, then, detect the optical density (OD) value of each well was detected with an enzyme marker and the cell viability of each group was calculated according to the formula: cell viability (%) = [(As-Ab)/(Ac-Ab)] * 100%.

## Detecting LIP levels

Cells were inoculated and treated according to the CCK8 method described above. At the end of the intervention the medium was removed and the wells were washed once with PBS. AM-calcein (1%) (YERSEN; Shanghai, China) was added and incubated for 30 min protected from light. After removing AM-calcein and washing two times with PBS, each well containing 100 μPBS was used to read the fluorescence values under an enzyme marker. The value of fluorescence was inversely proportional to the level of intracellular LIP.

## Glucose-stimulated insulin secretion test

The MIN6 cells were inoculated in 6-well plates and the interventions were performed separately according to the grouping. At the end of the intervention the old medium was removed and then added to Krebs-Ringer bicarbonate (KRB) buffer with 17.6 mmol/L glucose and incubation continued for 2 h in the cell incubator. Finally, the culture supernatant and cells were collected and insulin concentrations were assayed separately.

## Reactive oxygen testing

Cells were inoculated separately in 96-well plates (100 μl, $5 * 10^4$/ml) and the intervention was carried out according to the grouping. At the end of the intervention, we removed the old medium, loaded the probe (Beyotime; Shanghai, China) according to the instructions,

incubated for 20 min, then removed the probe solution, washed two times with PBS, and took fluorescence pictures of each group of cells under a fluorescent microscope after adding PBS.

## MDA testing

The cells were inoculated in 6-well plates at a density of $10 * 10^4$/ml and the intervention was carried out separately according to the grouping. After the intervention, the cells were collected in 1.5-ml Eppendorf (EP) tubes. The cells were then collected in 1.5-ml EP tube. Following the steps of the MDA assay kit (Solarbio; Beijing,China), the assay and calculations were carried out according to the manufacturer's instructions. The protein concentration was measured by the bicinchoninic acid (BCA) assay.

## Protein concentration assay and Western blotting

The cells were collected and lysed on ice for 30 min with the appropriate amount of radioimmunoprecipitation assay (RIPA) lysis buffer (Solarbio; Beijing, China) containing 1% protease inhibitor and phosphorylated protease inhibitor, and the lysed cells were further sonicated to ensure complete cell lysis. The supernatant after centrifugation is the protein sample. The protein concentration was determined by BCA and then denatured by adding protein loading buffer at 100 °C for 10 min in a metal bath to obtain samples for Western blotting. The subsequent experiments were carried out in strict accordance with the Western blot procedure.

## Method of blood collection

Tail-cutting method: the mouse tail tip, about 1 mm, was removed by a high temperature autoclave ophthalmic clipper. In order to avoid the influence of anesthetic on blood sugar, we did not use anesthetic.

Extirpation of eyeball for blood: first, we anesthetized mice by intraperitoneal injection of chloral hydrate (10%, 0.1 ml 10 g weight of mouse). To avoid hemolysis, we cut off the mouse's whiskers and then quickly removed the eyeballs to get enough blood samples. The mice were killed by cervical dislocation after blood collection.

## Insulin content testing

The insulin test was performed by an enzyme-linked immunosorbent assay (ELISA). The procedure was carried out according to the product instructions (Cloud-Clone; Wuhan, China). For *in vitro* experiments, we collected cells and cell culture medium supernatants for detection; for *in vivo* experiments, we collected the whole blood of the mice by the method of removing the eyeball to collect blood.

## Transmission electron microscopy for cellular ultrastructure

The treated cells were collected, added to a special electron microscope fixative and fixed for 24 h away from light. Pancreatic islets were obtained by first immobilizing the pancreatic tissue for at least 24 h using electron microscopic fixative, then refixing the pancreatic tissue into thin sections and hand-selecting the islet cell-rich pancreatic tissue under the microscope, then re-fixing the pancreatic tissue in the electron microscopic

solution, resin-embedding the samples and sectioning them. Staining was then performed using lead citrate and uranyl acetate, followed by transmission electron microscopy to visualize the cellular ultrastructure.

### Fasting blood glucose test in mice

The mice were first starved for 8 h. We took the blood by the broken tail method, swabbing the first drop at a time and waiting for the blood to regroup into a single drop for measurement.

### The oral glucose tolerance test (OGTT)

The mice were starved for 8 h prior to the experiment. Fasting blood glucose was measured first, and then gavaged at a dose of 1.5 g/kg of glucose, and blood glucose was measured in mice at 15, 30, 60, 90 and 120 min after gavage. The blood was collected as described above.

### Histopathological staining and analysis

We made paraffin sections of the pancreatic tissue, subjected the sections to conventional dewaxing and hydration and then stained the sections according to the different staining methods. The analysis of the images was carried out by ImageJ software.

### Statistical analysis

We analyzed the data with SPSS 20.0. The results were expressed as mean ± standard error of mean. $T$-test was used for the two groups of data conforming to normal distribution; one-way analysis of variance (ANOVA) was used for the three or more groups conforming to homogeneity of variance; otherwise, Kruskal-Wallis was used for statistical analysis. $P < 0.05$ was considered as the criterion for statistical significance.

## RESULTS

### The effect of iron overload on islet function in mice

The mice were divided into four groups: blank control group (the control group), deferasirox group (the DFX group), iron overload group (the IO group) and deferasirox treatment group (the IO+DFX group) to confirm the effect of iron overload on islet function. Mouse iron overload status was assessed by Prussian blue staining of pancreatic tissue. It can be observed from Fig. 1A that compared with the blank control and deferasirox groups, mice in the iron overload group had a large amount of ferritin deposition in the pancreatic tissue, suggesting that intraperitoneal injection of iron dextran resulted in severe pancreatic tissue iron overload. However, no significant improvement was observed after treatment with the iron chelator deferasirox. However, deferasirox therapy can relieve pancreatic iron deposition in patients with iron overload. Such results may be related to the duration of deferasirox treatment.

In order to evaluate the islet function of mice, fasting blood glucose, glucose tolerance, fasting insulin were measured, and insulin immunohistochemical staining was performed on pancreatic tissue of mice. The results of fasting blood glucose showed that compared with the control group, the fasting blood glucose of the IO group increased, with an average of 6.6 mmol/L; We also observed that gavage of deferasirox increased fasting blood

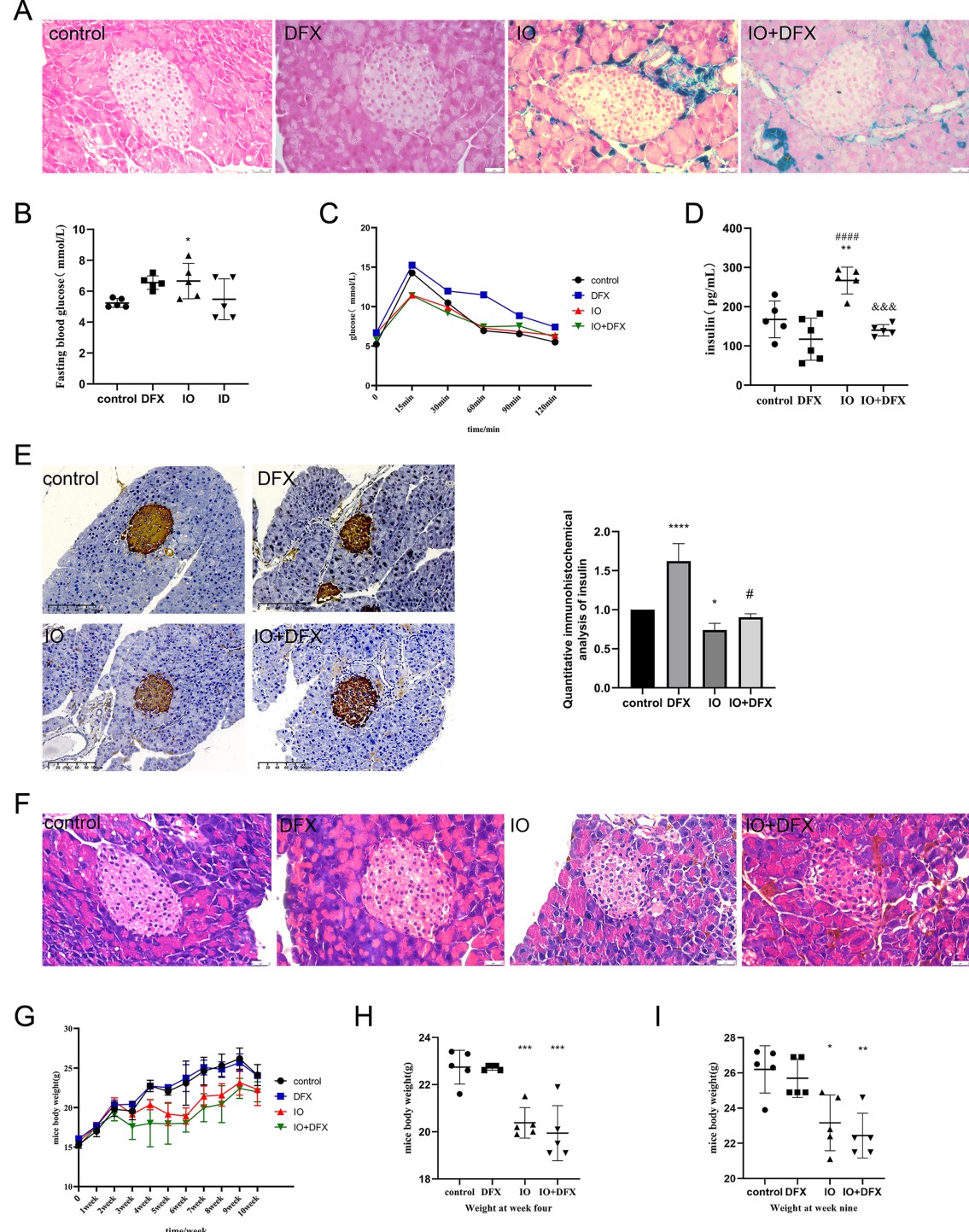

**Figure 1 Effects of iron overload on islet production in mice.** (A) Prussian blue staining of mouse pancreatic tissue, and the blue color indicated ferritin deposition. Scale bar: 25 μm; (B) fasting blood glucose of mice, *P < 0.05 *vs.* the control group; (C) OGTT in mice; (D) fasting insulin levels: **P < 0.01 *vs.* the control group, ####P < 0.0001 *vs.* the DFX group, &&&P < 0.001 *vs.* the IO group; (E) insulin immunohistochemical staining of mouse pancreatic tissue, brown color indicates positive expression area, scale scale: 100 μm. The mean optical density values of the positive areas were calculated by image J software to evaluate insulin expression, it is shown by the bar chart. *P < 0.05, ****P < 0.0001 *vs.* the control group, #P < 0.05 *vs.* the IO group; (F) HE staining of mouse pancreas, scale bar: 25 μm; (G, H, I) body weight of mice, *P < 0.05, **P < 0.01, ***P < 0.001 *vs.* the control group.

glucose by an average of 6.56 mmol/L in normal mice, but it improved the elevated fasting blood glucose induced by iron overload in mice (Fig. 1B). The results of glucose tolerance showed that the blood glucose of mice in each group reached the peak at 15 min after glucose gavage, and then slowly decreased. However, it was obviously observed that compared with the control group, the decrease rate of blood glucose in the DFX group was slowed down, especially after 30 min of glucose gavage; both IO and IO+DFX mice showed lower peak blood glucose levels (Fig. 1C). The results of fasting serum insulin showed that the serum insulin level of iron overload mice was 1.59 times higher than that of blank control mice. In contrast, this reduction was only 0.53 times as high in the deferasirox group as in the iron-overload group (Fig. 1D). These results suggest that the insulin sensitivity of iron overload mice is severely reduced, and deferasirox chelating iron treatment can improve insulin resistance. However, the results of anti-insulin staining of mouse islets showed that the insulin expression of mice in the iron overload group was significantly lower than that in the blank control group (Fig. 1E). These results suggest that iron overload not only leads to insulin resistance, but also directly impairs the ability of islet β cells to synthesize insulin.

In order to observe the pathological changes of pancreatic tissue, we also performed HE staining. The results showed that compared with the blank control group, the islets of iron overload group had obvious changes, mainly manifested as unclear boundary between islet cells and surrounding exocrine cells, and increased vacuoles in islet cells. Treatment with deferasirox resulted in a reduction in intracellular vacuoles (Fig. 1F). These results suggest that although the excess ferritin is mainly deposited in the exocrine part of pancreas, it still causes serious damage to islet cells.

In addition, we also observed that iron overload negatively affected the body weight gain of the mice. The weight of the mice was measured once a week, and the results showed that the weight of the mice in the iron dextran injection group increased slowly (Fig. 1G). The weight of the mice in the iron dextran injection group was significantly lower than that in the control group at the end of the model establishment (Figs. 1H and 1I).

These results suggest that iron overload directly impairs the ability of islet β cells to synthesize and secrete insulin, leading to abnormal glucose metabolism in iron overloaded mice.

## Iron overload induces ferroptosis in islet β cells

Based on the close relationship between iron metabolism and ferroptosis, we examined the role of ferroptosis in islet β cell dysfunction induced by iron overloading in min6 cells and isolated mouse islets.

*In vitro*, the intracellular free iron content and cell viability of min6 cells were evaluated to determine the conditions for establishing iron overload in min6 cells. The optimal intervention regimen was defined as better cell viability combined with higher LIP levels (more severe iron overload). As shown in Figs. 2A and 2B, we finally chose to intervene with FAC 640 μM for 48 h as the condition to establish iron overload in min6 cells, at which time the cell survival rate was 61.10%. We evaluated the effect of iron overload on the synthesis and secretion of insulin by min6 cells using glucose-stimulated insulin

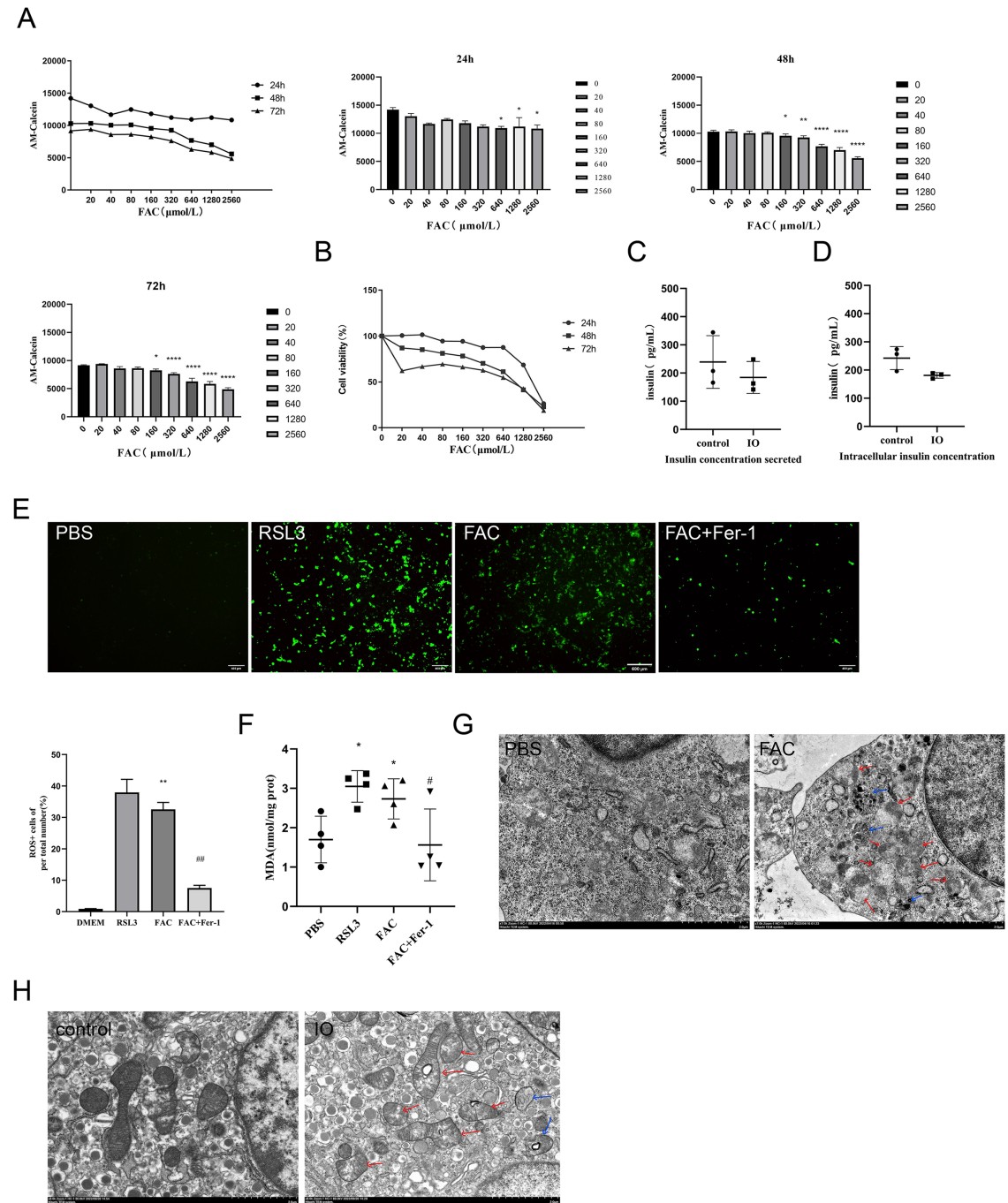

**Figure 2  Ferroptosis of islet β cells due to iron overload.** (A) Intracellular LIP levels in MIN6 cells, OD values were inversely proportional to LIP levels, as detailed in the Methods and Materials section. The LIP levels of each treatment concentration at 24 h, 48 h and 72 h were statistically analyzed, $^*P < 0.05$, $^{**}P < 0.01$, $^{****}P < 0.0001$ vs. the control group; (B) viability of MIN6 cells; (C and D) glucose-stimulated insulin secretion test in MIN6 cells; (E) ROS level in MIN6 cells, and green fluorescence showed positive cells. The proportion of positive cells was calculated and statistically analyzed by image J software, and the bar graph shows the quantification of positive cells. $^{**}P < 0.01$ vs. the control group, $^{##}P < 0.01$ vs. the FAC group; (F) MDA level in MIN6 cells; $^*P < 0.05$ vs. the control group, $^#P < 0.05$ vs. the FAC group; (G) the ultrastructure of MIN6 cells under the transmission electron microscope, blue arrows are autophagosomes, red arrows are shrunken mitochondria, and crista within mitochondria are reduced. Scale bar: 2 μm; (H) mouse islet β cells under the transmission electron microscope, blue arrows are autophagosomes, red arrows are mitochondria with significantly reduced cristae, and mitochondrial cristae within mitochondria is reduced. Scale bar: 2 μm.

secretion assay. The results showed that the insulin concentration of iron overload group was lower than that of control group. In the supernatant, the insulin concentration of iron overloading group was 0.76 times higher than that of control group ($P < 0.05$), as shown in Fig. 2C; The insulin concentration in the iron overloading group was 0.79 times higher than that in the control group ($P < 0.05$), as shown in Fig. 2D. Consistent with the *in vivo* results, iron overload impaired islet β cell function.

The occurrence of ferroptosis was determined by detecting the content of ROS and MDA in the cells, and observing the ultrastructure of the cells by transmission electron microscopy. In addition to the blank group, ferroptosis agonist and ferroptosis inhibitor groups were added as controls in this part of the validation. The results showed that ROS and MDA levels in the iron overload group were significantly higher than those in the blank group ($P < 0.01$; $P < 0.05$), and there was no significant difference compared with ferroptosis agonist group. The use of ferroptosis inhibitors significantly inhibited the formation of both products (Figs. 2E and 2F). These results suggest that iron overload may induce severe oxidative stress leading to the imbalance of intracellular REDOX state, resulting in the accumulation of large amounts of lipid peroxides and then induce ferroptosis.

Observation of cell ultrastructure is an important way to determine the mode of cell death. The ultrastructure of min6 cells was observed under the transmission electron microscope, and the results showed that compared with the blank group, the cell membrane of MIN6 cells in the iron overload group was incomplete, a large number of organelle debris and autophagosomes in the cytoplasm were observed, and the endoplasmic reticulum was swollen. Importantly, mitochondrial shrinkage and a marked reduction in mitochondrial cristae were observed (Fig. 2G). The ultrastructure of islet β cells isolated from pancreatic tissues of each group of mice was also observed and analyzed. Compared with the control group, the mitochondria of islet β cells in the iron overload group became smaller, the mitochondrial cristae decreased, and a large number of autophagosomes could be observed in the cytoplasm. Treatment with deferasirox increased intracellular autophagosomes, with more mitochondrial cristae relative to the iron-overload group (Fig. 2H).

These results confirmed that ferroptosis was involved in the process of iron overload-induced islet β cell dysfunction.

## Iron overload activates the ASK1/P-P38/CHOP pathway, leading to islet β cell dysfunction

To further verify whether iron overload-induced ferroptosis in islet β-cells is associated with the occurrence of CHOP-related ER stress, relevant indicators were validated. We divided min6 cells into groups: Control group (DMSO), deferasirox group (DFX), iron overload group (FAC), deferasirox treatment at low concentration group (FAC+DFX10), deferasirox treatment at medium concentration group (FAC+DFX20), and deferasirox treatment at high concentration group (FAC+DFX40); The protein expressions of ASK1, P-P38 and CHOP were detected by Western blot. The results showed that compared with the blank group, the protein expression levels of ASK1, P-P38, and CHOP in the iron

overload group were significantly increased, and deferasirox treatment could significantly inhibit the expression of the three proteins (Fig. 3A). To further clarify whether lalrox reversed the degree of lipid peroxidation in min6 cells, the protein expression of COX-2 was also detected. The results showed that the expression of COX-2 was reduced in the deferasirox treatment group compared with the iron overload group, especially in the medium and high concentration groups (Fig. 3B). These results suggest that iron overload-induced ferroptosis is induced by ASK1/P-P38/CHOP/COX-2 pathway in min6 cells, and deferasirox can inhibit endoplasmic reticulum stress induced by iron overload in a concentration-dependent manner.

Similarly, anti-ASK1, P-P38, CHOP, and COX-2 antibody staining of pancreatic tissue from mice showed that the expression of the four proteins was increased in the iron-overload group as compared with the control group, and the expression was inhibited by deferasirox treatment (Fig. 3C). This is consistent with our results in WB of min6 cells. In addition, ASK1 expression in the DFX group was also significantly increased compared with the blank control group. A relationship with side effects associated with deferasirox therapy was considered, but the cause remains to be determined.

## DISCUSSION

Since ferroptosis was proposed in 2012, research on the subject has grown exponentially (*Dixon et al., 2012*). In diabetic states, ferroptosis has been identified as a new determinant of islet β cell death (*Stancic et al., 2022*). However, few studies have directly linked iron overload, ferroptosis and islet β cell function in type 2 diabetes. This connection between iron overload and islet β cell function and type 2 diabetes was first observed in patients with thalassaemia and haemochromatosis, which is considered to be a major risk factor for glucose metabolism disorders (*Noetzli et al., 2012*; *Pepe et al., 2020*). Excessive ferritin deposition in the pancreas has also been observed in the course of type 2 diabetes (*Li et al., 2020*; *Zhao et al., 2022*), which may be an important mechanism. Therefore, we studied MIN6 cells and C57BL/6J mice exposed to iron overload as the only exposure factor with the aim of excluding other identified factors known to be risk factors for the induction of type 2 diabetes, to investigate the effect of iron overload on pancreatic islet β cell function and the occurrence of ferroptosis.

Our study renews the damaging effects of iron overload on glucose metabolic homeostasis, islet function and insulin secretion by islet β cells, and reveals the molecular mechanisms involved in endoplasmic reticulum stress associated with CHOP in resisting iron overload in islet β cells, where excessive activation induces ferroptosis in islet β cells. DFX administration improves the ability of islet β cells to secrete insulin and protects islet β cells from ferroptosis by inhibiting the relative protein expression of ASK1/P-P38/CHOP.

Consistent with previous findings, iron overload led to reduced insulin secretion from islet β cells (*Hamad et al., 2021*). Hyperinsulinaemia was observed in iron-overloaded mice (*Lee et al., 2021*). In this study, mice in the iron overload group (aged rats) had lower hourly fasting blood glucose levels due to hyperinsulinaemia. However, our results showed that mice in the iron overload group still had higher levels of fasting blood glucose despite

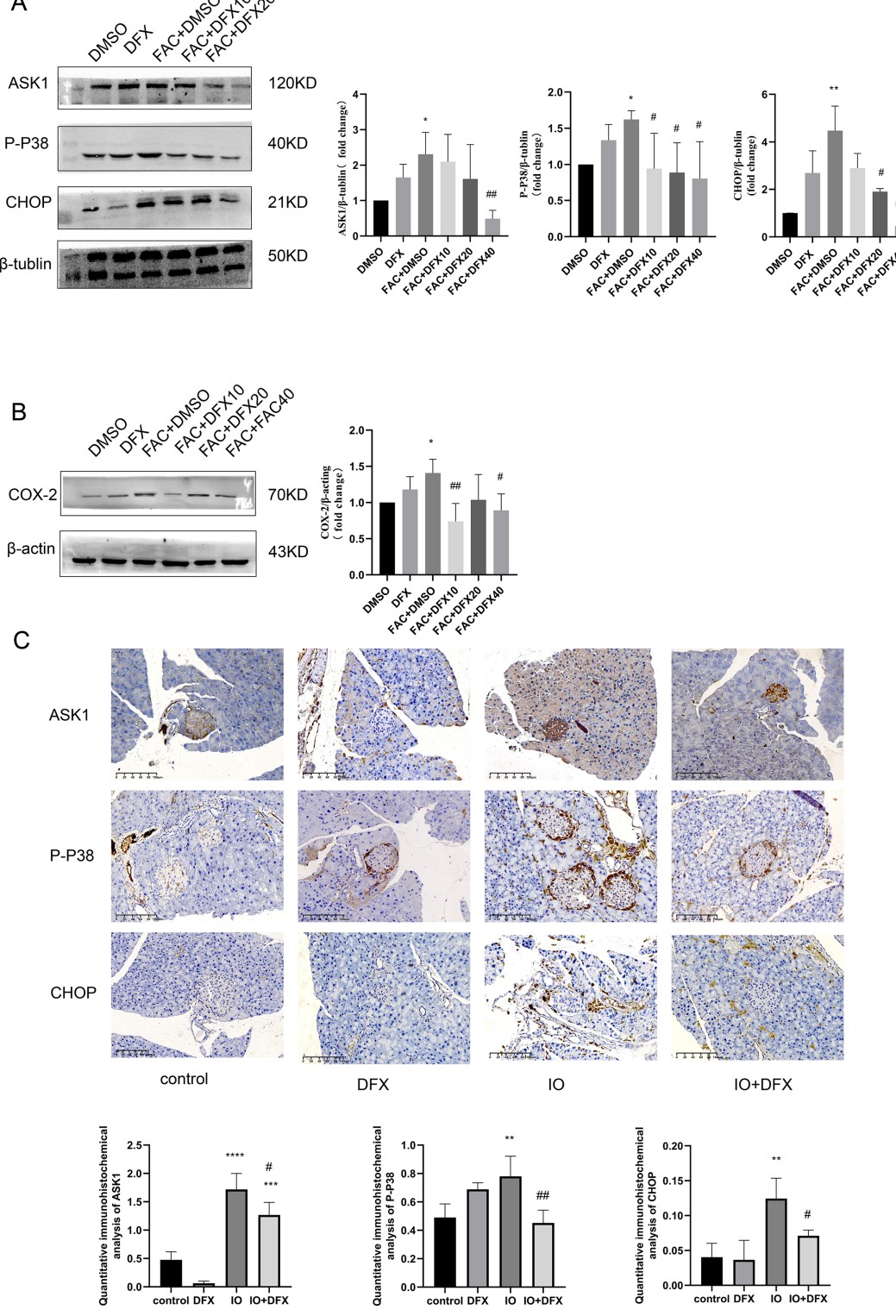

**Figure 3 Expression of ASK1, P-P38, CHOP, and COX-2 related proteins.** (A and B) Protein levels of ASK1, P-P38, CHOP, and COX-2 in the indicated groups were shown. Bar graph displays quantification of blots ($n = 3$). *$P < 0.05$, **$P < 0.01$, *vs.* the DMSO group; #$P < 0.05$, ##$P < 0.01$ *vs.* the FAC+DMSO group; (C) immunohistochemical staining for anti-ASK1, P-P38, and CHOP antibodies in pancreatic tissue, quantification of positively expressed areas in the cylinder ($N = 4$–5). **$P < 0.01$, ***$P < 0.001$, ****$P < 0.0001$ *vs.* the control group; #$P < 0.05$, ##$P < 0.01$ *vs.* the IO group.
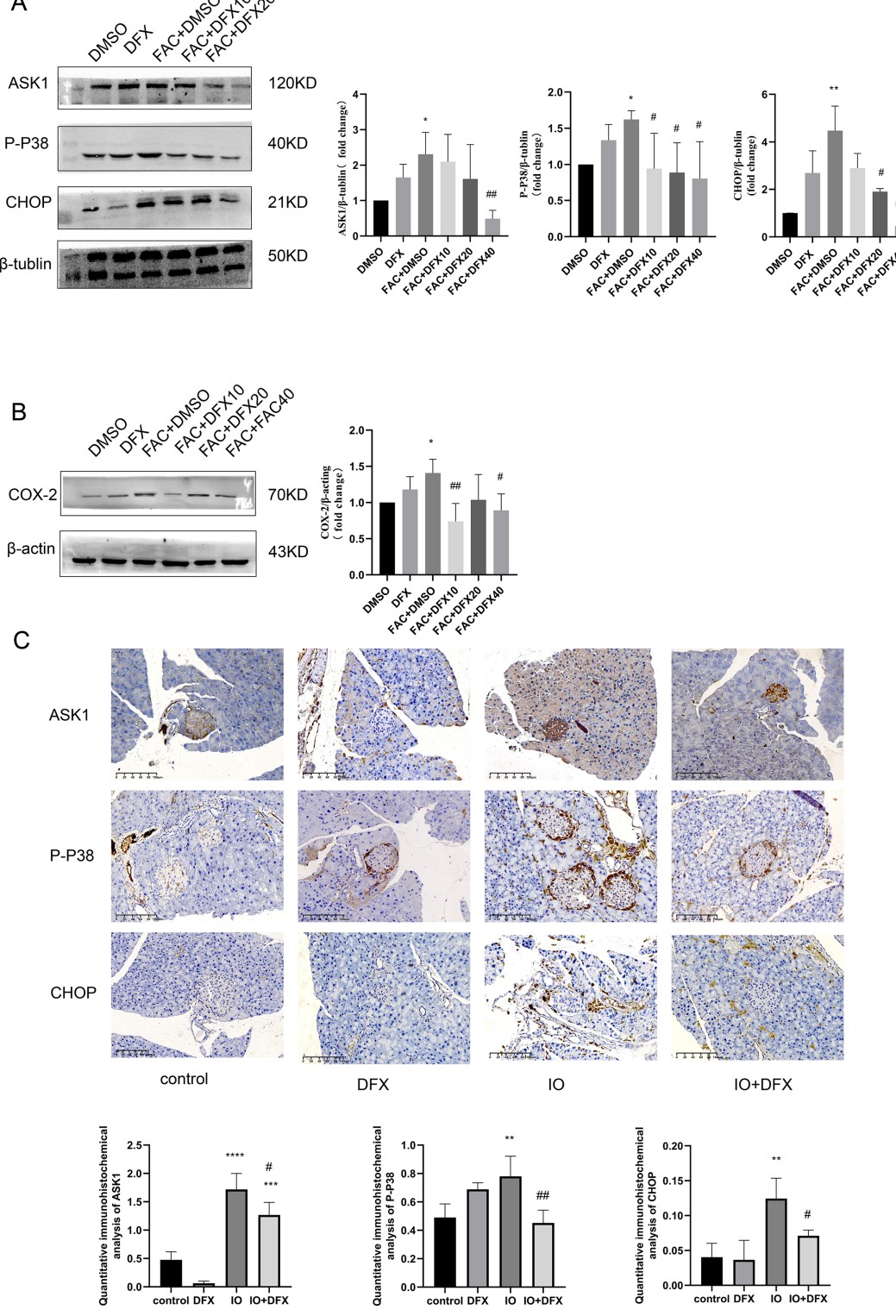

the hyperinsulinemia. This is consistent with the results of most clinical studies (*Tangvarasittichai et al., 2013*; *Yassin et al., 2018*; *Mahmoud, Khodeary & Farhan, 2021*). The reason for this the higher levels of fasting blood glucose may be difference in the degree of iron overload leading to different degrees of insulin resistance. Furthermore, in the 2 h OGTT test, we did not observe the classical profile of insulin resistance in iron-overloaded mice; instead, iron-overloaded mice exhibited a lower peak blood glucose but still required 60 min postgavage to return to normal blood glucose levels, suggesting that iron-overloaded mice may have a defect in insulin synthesis. This defect was confirmed by the results of anti-insulin staining of mouse pancreatic tissue. The lower peak blood glucose may be related to the damaged intestinal epithelium and reduced glucose absorption capacity due to iron overload (*Zhang et al., 2022b*). Instead, we observed a more pronounced insulin resistance profile in DFX control mice. Previous studies have shown that DFX has significant hepatotoxicity and nephrotoxicity (*Kontoghiorghe & Kontoghiorghes, 2016*; *Adramerina et al., 2022*), which could be the reason why we observed an increase in blood glucose, pathological islet changes and increased expression of P-P38 and ASK1 in the DFX group. However, to the best of our knowledge, there is no further research evidence of a damaging effect of DFX on fasting blood glucose or islets. Studies have shown that the use of DFX or desferrioxamine in the absence of iron overload can induce apoptosis by regulating iron metabolism and activating the MAPK signalling pathway (*Xue et al., 2021*).

Peroxidation of polyunsaturated phospholipids, accumulation of redox-active iron and loss of lipid peroxidation repair capacity are measures of the extent to which ferroptosis occurs (*Feng et al., 2020*). To verify the occurrence of ferroptosis, we examined lipid ROS and MDA levels and the protein expression levels of COX-2. As with most islet β cells that undergo ferroptosis, we detected high levels of ROS and MDA in iron-overloaded MIN6 cells and significantly increased protein expression of COX-2 (*Kitabayashi et al., 2022*; *Hong et al., 2022*; *Krümmel et al., 2021*). We suggested that the accumulation of excess lipid peroxides leads to a relative decrease in the antioxidant capacity of MIN6 and may be the direct cause of ferroptosis in islet β cells due to iron overload.

Islet β cells are estimated to produce approximately one million insulin molecules per minute to maintain normal blood glucose levels. The secretory pathway synthesizes proteins mainly in the endoplasmic reticulum, so the protein folding mechanism in the endoplasmic reticulum is important for secretory cells such as islet β cell (*Liu et al., 2018*; *Arunagiri et al., 2019*). CHOP is an endoplasmic reticulum stress-related factor, encoded by GADD153, which is also thought to be an important signalling transcription factor in the regulation of insulin synthesis. In db/db leptin receptor-deficient mice, chronic endoplasmic reticulum stress leads to CHOP-dependent islet β cell death; knockdown of the Chop gene alleviates endoplasmic reticulum stress and oxidative stress to reduce islet β cell death, thereby reversing diabetic hyperglycaemia in these db/db mice (*Song et al., 2008*). Chronic endoplasmic reticulum stress has been shown in numerous studies to clear impaired islet β cells mainly through induction of apoptosis (*Hetz, 2012*; *Eizirik, Pasquali & Cnop, 2020*; *Jin et al., 2022*; *Zhang et al., 2020c*). In recent years, ER stress-related activation of the PERK pathway has nevertheless been shown to leads to islet β cell

dysfunction associated with cellular ferroptosis (Zhang et al., 2022a). Our study demonstrates that iron overload leads to impaired islet β cell function by inducing endoplasmic reticulum stress associated with CHOP, while ferroptosis is observed in islet β cells during this process.

In addition, autophagy, apoptosis, necrosis and other forms of cell death are also associated with islet β cell dysfunction and ferroptosis. For example, autophagy of ferritin, increased intracellular free iron content and the accumulation of reactive oxygen species all play a key role in the development of ferroptosis (Gao et al., 2016). We found large numbers of autophagic lysosomes in iron-overloaded islet β cells, however the connection between autophagic lysosomes and ferroptosis needs to be verified in more studies.

In conclusion, this study elucidates the role of iron overload as the sole exposure factor leading to islet β cell dysfunction, suggesting a direct damaging effect of iron overload on islet β cells. ER stress is also suggested to be involved in this process, and inhibiting the accumulation of misfolded proteins may suppress the damage to islet β cells by iron overload.

# ACKNOWLEDGEMENTS

We thank Professor Xia Ning for her guidance on this manuscript.

## Funding

This work was supported by the Natural Science Foundation of Guangxi: 2017GXNSFDA198010. The funders had no role in study design, data collection and analysis, decision to publish, or preparation of the manuscript.

## Grant Disclosures

The following grant information was disclosed by the authors:
Natural Science Foundation of Guangxi: 2017GXNSFDA198010.

## Competing Interests

The authors declare that they have no competing interests.

## Author Contributions

- Ling Deng conceived and designed the experiments, performed the experiments, analyzed the data, prepared figures and/or tables, and approved the final draft.
- Man-Qiu Mo conceived and designed the experiments, performed the experiments, prepared figures and/or tables, and approved the final draft.
- Jinling Zhong performed the experiments, prepared figures and/or tables, and approved the final draft.
- Zhengming Li analyzed the data, authored or reviewed drafts of the article, and approved the final draft.
- Guoqiao Li analyzed the data, authored or reviewed drafts of the article, and approved the final draft.

- Yuzhen Liang conceived and designed the experiments, authored or reviewed drafts of the article, and approved the final draft.

## Animal Ethics

The following information was supplied relating to ethical approvals (*i.e.*, approving body and any reference numbers):

All animal procedures were approved by the People's Republic of China 2021 Guide for Euthanasia of Laboratory Animals (GB/T39760-2021), the 2018 Guide for Ethical Review of Laboratory Animal Welfare (GB/T35892-2018), and the Guangxi Medical University Laboratory Animal Ethics Review Body (Project Licences: 20211007)

## Data Availability

The raw measurements are available in the Supplemental Files.

## Supplemental Information

Supplemental information for this article can be found online at http://dx.doi.org/10.7717/peerj.15206#supplemental-information.

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
