# Peer review of "Iron overload induces islet β cell ferroptosis by activating ASK1/P-P38/CHOP signaling pathway"

_PeerJ, doi:10.7717/peerj.15206_

## Round 0.1 · original submission · Major Revisions

Dear Dr. Liang,

Your article - Basic research on iron overload-induced pancreatic β cells dysfunction - that it requires a Major Revisions.

The reviewer suggested comments should be taken into consideration.

Waiting to receive your revised version of the manuscript.

Kind regards,

Reviewer 1 ·

Basic reporting

The authors studied iron overloading induced pancreatic b-cells dysfunction in MIN6 cells and mouse. Although there are several major and minor flaws, the paper has a clear logic and possible explanations on open questions. I would recommend some revisions.

Major issues:

1: The title of this manuscript should be refined to give a clear conclusion instead of “basic research on…” which currently makes the study look not novel and important.

2: It is better treat mouse models with gradient dose of iron dextrose and DFX, which needs to be incorporated for comparison.

3: Basically, the writing goes with the logic flow. It is not necessary to split the whole story to cellular experiments and animal experiments, and thus the overall architecture of this manuscript could be optimized.

4: Line 280-281, the data should be provided as supplementary figures to conclude exogenous iron do not have a hepcidin antimicrobial peptide (HAMP) gene defect.

5: Figure 2C is hard for the audience to get clear information and needs to be polished. For instance, different symptoms in Line 242-246 should be marked respectively.

Minor issues
1: Line 24, change “from” to “using”.

2: Line 72, “the other hand” should be “On the other hand”

3: Line 194-199, Method of Blood Collection should be above OGTT, because the author simplifies the blood collection as “The blood was collected as described above.”

4: Line 230-231, please give the reference literatures of the last sentence.

5: Line 357, change “may difference” to “may be different”

6: Line 330, change “vitro experiments” to “ in vitro experiments”.

Experimental design

Please see comments in Basic reporting

Validity of the findings

Please see comments in Basic reporting

Reviewer 2 ·

Basic reporting

The hypothesis of this study is well described and tested with the appropriate in vivo and in vitro methods accordingly. The results of this study give significant support to the hypothesis proposed. It is clear that all experiments were performed carefully and analyzed and presented well. Although there are minor grammatical errors, the sentences are clear and understandable. All figures are presented well, and the captions of figures are descriptive.
I suggest that this study deserves to be considered for publication revision by the comments given below:
1. Title is somehow general. However, “basic research” is a very broad term though this study investigates certain aspects of the dysfunction of iron overload pancreatic cells. Therefore, the title of the page should be revised to cover the content of this study.
2. 3. Line 26: MIN6 is written with lover cases; please check the manuscript thoroughly to be consistent with the usage of such abbreviations.

Experimental design

Study is well-designed to test the hypothesis of this study. However, the doses of compounds given to the mouse groups and treatment durations are given, but the principles of using these dosages and exposure duration need to be explained.

Validity of the findings

All the findings of this study were obtained using appropriate methods to test the hypothesis of the proposed. The reliability of the findings was determined by applying statistical tests when necessary. The findings are supported by the available literature data and the results are discussed accordingly.

Reviewer 3 ·

Basic reporting

In this research article entitled “Basic research on iron overload-induced pancreatic β cells dysfunction ”, the authors (Deng et al,) checked if the iron overload may play a role in driving glucose metabolism abnormalities by inducing endoplasmic reticulum stress that mediates ferroptosis in pancreatic β-cells .
Hereafter, some points that should be taken into account before processing to the next steps.
Comments to the authors:
- The use of “basic research” in the title of the manuscript is not relevant.
- The part C of figure 2 is not clear. Some modifications on the contrast of the SEM slides should be realized to improve the figure, which requires already a scale bar.
- The scale bars of the figure 6 are not clear.
- The authors should emphasize the reason behind the selection of the used doses. More clarifications are needed.
- Check figure 3 legend. Several boxes exist within the abbreviations, such as lines 1 and 3 of the figure 3 caption.
- The authors should implement statistical analyses for the figure 1 data. Furthermore, the use of symbols, such as asterisks, is recommended.

Experimental design

Good expérimental design

Validity of the findings

Fine

---

## Round 0.2 · accepted · Accept

Dear Dr. Liang,

Your manuscript - Iron overload induces islet β cell ferroptosis by activating ASK1/P-P38/CHOP signaling pathway - has been Accepted.

kind regards,

Reviewer 1 ·

Basic reporting

The authors have resolved my concerns, and I think the current version is suitable for publication.

Experimental design

NA

Validity of the findings

NA